

# Comparative study of SPI success factors in global and in-house environment for large-scale software companies

Javed Iqbal[1], Haris Jibran[2], Ahmad Sami Al-Shamayleh[3], Fakhar Abbas[4], Adnan Akhunzada[5], Salman Z. Alharthi[6] and Abdullah Gani[7]

[1] Department of Computer Science, COMSATS University, Islamabad, Pakistan
[2] Computer Science Department, COMSATS University, Islamabad, Pakistan
[3] Department of Networks & Cybersecurity, Faculty of Information Technology, Al-Ahliyya Amman University, Amman, Jordan
[4] Centre for Trusted Internet and Community, National University of Singapore, Singapore, Singapore
[5] College of Computing & IT, University of Doha for Science & Technology, Doha, Qatar
[6] Department of Information System, College of Computers and Information Systems, Umm AL-Qura University, Al-lith, Saudi Arabia
[7] Faculty of Computer Science and Information Technology, University of Malaya, Kuala Lumpur, Malaysia

Corresponding author
Javed Iqbal,
javedkhushi@hotmail.com

## ABSTRACT

**Background**. Software process improvement (SPI) is an indispensable phenomenon in the evolution of a software development company that adopts global software development (GSD) or in-house development. Several software development companies do not only adhere to in-house development but also go for the GSD paradigm. Both development approaches are of paramount significance because of their respective advantages. Many studies have been conducted to find the SPI success factors in the case of companies that opt for in-house development. Still, less attention has been paid to the SPI success factors in the case of the GSD environment for large-scale software companies. Factors that contribute to the SPI success of small as well as medium-sized companies have been identified, but large-scale companies have still been overlooked. The research aims to identify the success factors of SPI for both development approaches (GSD and in-house) in the case of large-scale software companies.

**Methods**. Two systematic literature reviews have been performed. An industrial survey has been conducted to detect additional SPI success factors for both development environments. In the subsequent step, a comparison has been made to find similar SPI success factors in both development environments. Lastly, another industrial survey is conducted to compare the common SPI success factors of GSD and in-house software development, in the case of large-scale companies, to divulge which SPI success factor carries more value in which development environment. For this reason, parametric (Pearson correlation) and non-parametric (Kendall's Tau correlation and the Spearman correlation) tests have been performed.

**Results**. The 17 common SPI factors have been identified. The pinpointed common success factors expedite and contribute to SPI in both environments in the case of large-scale companies.

# INTRODUCTION

One of the software development mechanisms known to the software realm is global software development (GSD). This practice is based on the idea of outsourcing the work to a third-party vendor. The vendor may be located anywhere in the world (*Akbar et al., 2019*). GSD is an agreement between a client and a vendor, according to which the client hires the vendor to develop a particular software based on the defined requirements' specifications (*Ullah Khan, Niazi & Ahmad, 2010*). The concept of the GSD has been around since the 1970s. At that time, this concept was known as "Contract Programming", where a small part of the software development would be handed over to a third-party (*Khan et al., 2017a*). Despite the diverse advantages of GSD, many organizations still choose in-house software development for their daily development needs. They prefer developing the software systems by utilizing the workforce from within the organization as opposed to hiring a third-party. Such companies have successfully completed the projects and they want to continue with the same team instead of hiring new vendors from outside.

Both these software development approaches have their pros and cons. In any given case, a company decides what approach to adopt based on the company's preferences. Whether a company chooses the GSD approach or the in-house software development approach, there are certain risks associated with it. That is where software process improvement (SPI) comes into play. The competence of an overall process is established by looking at the performance of its designated sub-processes in human-intensive activities like software development. Most of the causes for poor quality and productivity are thus managed or removed as the capability of each sub-process is improved (*Humphrey, 1993*). SPI approaches are presented in connection to or as part of some well-known process improvement frameworks. The CMMI (Capability Maturity Model Integration), SPIRE, SATASPIN, PRISMS, MESOPYME, MoProSoft, and MPS are well-known SPI frameworks that specify SPI techniques (*Aysolmaz & Demirörs, 2011*). The definition of the SPI, as proposed by *Hansen, Rose & Tjørnehøj (2004)* is: "SPI is an applied academic field rooted in the software engineering and information systems disciplines. It deals with the professional management of software firms, and the improvement of their practices, displaying a managerial focus rather than dealing directly with the techniques that are used to write software". SPI has been adopted by several renowned large companies such as Alcatel, Ericsson, Hughes Aircraft, and Motorola (*Pekki, 2016*).

GSD is one of the most common modes of software development in the modern era. Since there are no pre-defined techniques to improve GSD based on SPI in the case of large companies, therefore, the companies that make use of GSD need to create their guidelines and follow them to be successful in the international market. Creating effective guidelines also allows such companies to deliver better products to clients and to bring innovative solutions to the market (*Khan et al., 2019*; *Zhang et al., 2019*; *Roman et al., 2022*).

Success factors contribute towards a well-executed software development process. These success factors are applicable in the case of GSD and in-house software development. Generally, there are four types of companies based on the number of employees and turnover. These categories include micro, small, medium, and large companies. Large

enterprises have more than 250 employees, with a turnover of larger than 50 million euros (*Fernández et al., 2019*). This research work aims at identifying the common SPI success factors for GSD and in-house software development in the case of large-scale software development organizations. The next step is to find the relative importance of each common SPI success factor, for GSD and in-house software development, in the case of large-scale software development organizations. For this purpose, the following research questions have been designed:

RQ1. Which are software process improvement success factors for global software development in the case of large-scale software development organizations?

RQ2. Which are software process improvement success factors for in-house software development in the case of large-scale software development organizations?

RQ3. Which are the additional software process improvement success factors, for global software development and in-house software development, in the case of large-scale software development organizations?

RQ4. Which are the common software process improvement success factors, for global software development and in-house software development, in the case of large-scale software development organizations?

RQ5. What is the relative importance of each common software process improvement success factor, for global software development and in-house software development, in the case of large-scale software development organizations?

## RELATED WORK

*Lee, Shiue & Chen (2016)* has examined the impact of top management and organizational culture on knowledge sharing for SPI using a survey research approach and has used the partial least squares technique to analyze the samples. *Niazi (2015)* has performed a comparative study of the success factors of SPI with previously identified success factors. *Khan et al. (2017b)* identified human-related factors that positively affect SPI in GSD organizations. Moreover, *Anwer et al. (2019)* did a comparison of challenges for requirement change management (RCM) in GSD and in-house environments. Coding, testing, planning, and packaging are all examples of software development processes. These processes can be enhanced for better quality, on-time, and within budget product delivery; and the activity that improves these processes is known as SPI (*Farooq et al., 2021*). The quality of a product is largely determined by the methods that a company employs to develop software (*Söylemez & Tarhan, 2018*), as well as the competency and maturity of these methods and techniques (*Kabitimer, Midekso & Machado, 2018*). The processes to be improved must be chosen based on the corporate environment, culture, and priorities. More importantly, the effectiveness of an SPI program is determined by the business results it can produce; therefore, it must relate to the organization's business objectives (*Vasconcellos et al., 2017*).

The alignment of SPI and business goals is a significant aspect of an SPI initiative's success. *Bayona, Calvo-Manzano & San Feliu (2012)*, for example, quote alignment with business strategy and goals as one of 16 crucial variables for process improvement

implementation success, while Dyba's research (*Dyba, 2000*; *Dybå, 2003*; *Dyba, 2005*) emphasizes business orientation as one of the main supporting aspects for SPI. Each modification of a software process might be considered a strategic decision. This is called strategic alignment of SPI because aligning SPI with business goals necessitates modifications in a software process and is a strategic choice (*Münch et al., 2012*). Organizational factors influence SPI implementation, and numerous studies have looked into the important factors for SPI achievement (*Lee & Chen, 2019*). For example, *Niazi, Wilson & Zowghi (2006)* have highlighted seven variables that are deemed crucial for successful SPI implementation. *Sulayman et al. (2014)* did a thorough analysis of current SPI research and developed an integrated framework for small and medium enterprises (SMEs), that included 18 categories of SPI success variables. Small businesses sometimes lack the knowledge to seek and adapt process improvement best practices from a variety of frameworks to their own needs. Finally, small businesses are frequently on the lookout for low-cost evaluation or certification programs that would help them gain recognition (*Laporte & O'Connor, 2017*).

Although SMEs' role in the economy and job creation matters, but they confront a variety of obstacles and roadblocks on the way to SPI (*Basri et al., 2019*). Software development SMEs are unable to incorporate SPI into their operations because of time constraints, insufficient resources, and a lack of support. Furthermore, some studies (*Almomani et al., 2015*; *Nasir, Ahmad & Hassan, 2008*) have revealed that SPI operations are underutilized in small and medium software development organizations. The challenges and advancements in the field of software engineering have prompted the creation of a variety of SPI frameworks, ranging from traditional plan-driven frameworks to modern lean agile-based frameworks. Top-down models CMMI, ISO/IEC 330XX, and ISO/IEC 29110 are prescriptive models based on a collection of best practices that have been proven successful (*Sharma & Sangal, 2019*). It was found that about 40 percent of the business processes failed due to inefficient RCM methodologies. Over the years, various models and mechanisms have been adopted to solve issues arising from RCM (*Ahmad, Khan & Khan, 2021*). A study focused on the application of the Capability Maturity Model (CMM) by involving a team of fewer than 10 people. This was observed that the importance of team management is as important as process management (*Wongsai, Siddoo & Wetprasit, 2015*).

From this wide spectrum of studies, this can be concluded that no study focuses on the identification of SPI success factors in the case of such large-scale software development companies that opt for GSD as well as in-house development. The identification of SPI success factors ensures SPI implementation and SPI has emerged as the main strategy for enhancing software quality, as well as staff and client satisfaction (*Herranz et al., 2019*; *Anastassiu & Santos, 2020*; *Khan et al., 2017c*).

## METHODOLOGY

We have carried out two systematic literature reviews (SLRs) for this study and have performed two questionnaire surveys. In the first step, we have employed the SLR to find out the SPI success factors for GSD in the case of large-scale software development

organizations (RQ1). In the next step, we have completed a second SLR to find out the SPI success factors for the in-house context in the case of large-scale software development organizations (RQ2). Based on the first and second steps, we have developed a questionnaire survey to find out if there are additional SPI success factors for both contexts in the industry or not (RQ3). After this, to answer RQ4, we have spotted similar SPI factors in both development environments, and the strength of each common SPI success factor has been deduced by another questionnaire survey to see which SPI success factor is more important in which development environment (RQ5). Figure 1 depicts the methodology employed for this study.

## Systematic literature review

The SLR is among the most widely adopted methodologies when it comes to the use of evidence-based software engineering. One of the key points of an SLR is that it is a well-planned and systemically executed scheme. In this research, we have followed *Kitchenham & Charters (2007)* guidelines to execute the SLR process. This SLR process consists of three main steps: (1) defining the protocol, (2) conducting the protocol, (3) reviewing the protocol. In the first step, the protocol contains these elements: (1) research question's identification, (2) search strategy, (3) study selection, (4) quality assessment, (5) extraction of data and synthesis. The first element *i.e.,* identification of the research question has already been mentioned in the introduction section; search strategy, study selection criteria, and quality assessment are described in section A whereas extraction and synthesis of data are included in results and discussion. This research has been conducted by one student and two academic staff members to avoid biases.

### *Search strategy*

For the search strategy, four steps have been designed.

    1. Construct search terms by identifying keywords.

    2. The next step is finding the synonyms of the keywords. We used synonyms that have already been used by the researchers to assure validity (*Anwer et al., 2019*; *Mas et al., 2012*).

    3. We used Boolean operators to connect our terms. The operator "OR" is used to connect the synonyms of the keyword and "AND" is used to connect the keywords. In the case of in-house SPI success factors, we used "NOT" to remove the results related to GSD.

    The strings for various key terms are:

Success factors: "Success factors" OR "Critical success factors" OR "important success factors" OR "CSF" OR "key factors" OR "human factors" OR "social factors".

Software process improvement: "Software process improvement" OR "SPI" OR "process deployment" OR "software process implementation".

Global software development: "Global software development" OR "Global software engineering" OR "GSD" OR "Offshore software development" OR "Distributed software development" OR "Offshore outsourcing" OR "GSE" OR " DSD".

For the success factors of SPI in the GSD context following search string has been designed: ("Global software development" OR "Global software engineering" OR "GSD" OR "Offshore software development" OR "Distributed software development" OR

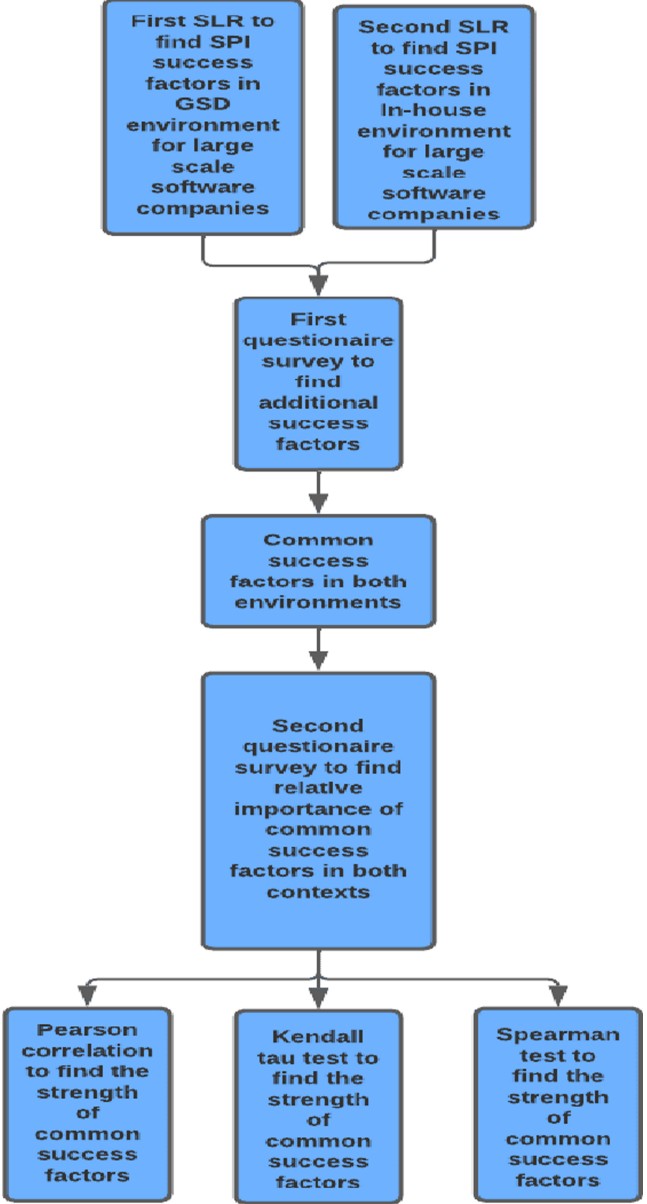

**Figure 1  Research methodology.**

"Offshore outsourcing" OR "GSE" OR " DSD") AND ("Success factors" OR "Critical success factors" OR "important success factors" OR "CSF" OR "key factors" OR "human factors" OR "social factors") AND ("Software process improvement" OR "SPI" OR "process deployment" OR "software process implementation").

For the success factors of SPI in an in-house context following search string has been designed: ("Success factors" OR "Critical success factors" OR "important success factors" OR "CSF" OR "key factors" OR "human factors" OR "social factors") AND ("Software process improvement" OR "SPI" OR "process deployment" OR "software

**Table 1  Electronic database searched for SLRs.**

| Sr. # | Database | Link |
|---|---|---|
| 1 | IEEE | https://ieeexplore.ieee.org/Xplore/home.jsp |
| 2 | ACM | https://dl.acm.org/ |
| 3 | Springer | https://link.springer.com/ |
| 4 | Science Direct | https://www.sciencedirect.com/ |
| 5 | Wiley | https://onlinelibrary.wiley.com/ |

**Table 2  Quality assessment questions.**

| Sr. # | Questions | Possible answers |
|---|---|---|
| 1 | Are the aims of the research clearly stated in the paper? | Yes = 1, No = 0 |
| 2 | Is the technique well-presented and justified? | Yes = 1, No = 0 |
| 3 | Is the methodology appropriate and applied adequately? | Yes = 1, No = 0 |
| 4 | Is the paper well referenced? | Yes = 1, No = 0 |

process implementation") NOT ("Global software development" OR "Global software engineering" OR "GSD" OR "Offshore software development" OR "Distributed software development" OR "Offshore outsourcing" OR "GSE" OR " DSD").

4. Resources searched in this study include specific research databases, journals, and conference proceedings. The five electronic academic databases were used to search for relevant primary studies. Table 1 provides details of the electronic databases.

### Study selection criteria

We have used inclusion and exclusion criteria to select the primary studies from the databases after applying the search strings. The criteria for the inclusion and exclusion of primary studies are as follows:

(1) INCLUSION CRITERIA

- Publications directly related to our research questions.
- Research paper from 2010-2021.
- Conference, journal, and magazine papers.
- Papers in the English language only.
- Papers length $\geq 3$.

(2) EXCLUSION CRITERIA

- Duplicate publications.
- Papers that were not in the English language.
- Publications, without bibliographic information.

### Quality assessment

The questions that have been used to evaluate the research quality are shown in Table 2.

### Data collection *via* questionnaire surveys

An empirical survey is an appropriate research methodology for collecting qualitative and quantitative data from a large group of participants by using techniques such as a questionnaire survey or interview (*Niazi, Babar & Verner, 2010*). To perform the questionnaire survey, we have used guidelines provided by *Kitchenham & Pfleeger (2008)*. In this research work, online questionnaire surveys have been used because the traditional survey approach has many problems (*Misro et al., 2014*). During the first SLR, we have identified the 35 SPI success factors for GSD in the case of large-scale software development organizations. Similarly, during the second SLR, we have identified the 33 SPI success factors for the in-house context in the case of large-scale software development organizations. Based on the results of both SLRs, we have developed the first questionnaire survey to ask industry professionals about the additional SPI success factors for both contexts, in the case of large-scale software development organizations, according to their own experience.

The questionnaire survey has three sections: section-I is related to demographics, section-II contains the list of SPI success factors in case of GSD and request to mention additional SPI access factors, and section-III presents the list of SPI success factors in case of in-house development and request to mention additional SPI access factors. Through prior contact *via* email and telephone, the participants were informed that the data would only be accessible to the research team and only be used for research purposes.

The second questionnaire survey aims to find out the relative importance of common SPI success factors in both contexts. The questionnaire for the second survey has two sections. The first section is to obtain demographics whereas the second section contains 17 common SPI success factors and a Likert scale to measure the relative importance of success factors. The survey employs a 5-point Likert scale that ranges from 1 to 5. The '1' indicates less importance and '5' indicates more importance.

The questionnaire surveys were first tested through a pilot study involving five professionals from different organizations. Based on the results of the pilot study, the final versions of the questionnaire surveys were developed.

### *First questionnaire survey*

The details of the first questionnaire survey are:

(a) RESPONDENTS

We sent the questionnaire survey to the professionals working in the industry. The questionnaire was distributed to professionals who were working in GSD and in-house development environments. This survey is a cross-sectional study.

(b) QUESTIONNAIRE FORMAT

The basic aim of this first questionnaire survey is to find additional success factors in both development environments. The questionnaire survey includes the basic information of the respondents such as email, position/job title, experience in years, and the primary business function of the company (GSD or in-house). The list of success factors generated from the SLR was also included in the survey so that the respondents may not enter the success factor again.

(c) SAMPLING AND POPULATION

Various methods for collecting samples from populations have been developed by researchers. In different situations, different sampling methods are acceptable for different objectives (*Baltes & Ralph, 2022*). The method used in this research is stratified sampling. In Stratified Random sampling, the population is partitioned into smaller groups that are called strata. Then a sample is drawn from each stratum (*Kitchenham & Pfleeger, 2008*). We invited 50 professionals to participate in the survey *via* Google Forms.

Our respondents have a minimum experience of 3 years and have been working as developers, software engineers, analysts, team leaders, project managers, and AI training engineers.

(d) RESPONSE RATE

We invited 50 professionals; in return, we got 26 correct responses.

***Second questionnaire survey***

The details of the second questionnaire survey are:

(a) RESPONDENTS

In the second survey, we sent the questionnaire survey to the professionals working in the industry. The questionnaire was distributed to professionals who were working in GSD and in-house development environments. The second survey is a cross-sectional study.

(b) QUESTIONNAIRE FORMAT

The basic aim of this questionnaire survey was to find out the relative importance of each common SPI success factor. The first part of this questionnaire includes email, experience, position or job title, and the primary business function of the company. In the second part, a Likert scale was used to find the importance of each common SPI success factor from 1 (which shows less importance) to 5 (which shows more importance).

(c) SAMPLING AND POPULATION

Various methods for collecting samples from populations have been developed by researchers. In different situations, different sampling methods are acceptable for different objectives (*Baltes & Ralph, 2022*). The method used in this research is stratified sampling. In Stratified Random sampling, the population is partitioned into smaller groups that are called strata. Then a sample is drawn from each stratum (*Kitchenham & Pfleeger, 2008*). We invited 50 professionals to participate in the survey *via* Google Forms.

The respondents of the survey have experience of 1 to 5 years and have been working as software engineers, software quality assurance managers, data annotators, product designers, AI engineers, project managers, and product coordinators.

(d) RESPONSE RATE

We sent this survey to 50 professionals and got responses from 29 people.

## Statistical tests

Firstly, we have conducted the parametric correlational analysis (Pearson) (*Obilor & Amadi, 2018*) which is not appropriate for a small sample size, even though we have used it to achieve results similar to non-parametric tests. The non-parametric correlation (Kendall's Tau and Spearman) tests are suitable for the analysis of the results of a small

sample size (*Fagerl, 2012*). Kendall's Tau test is used because it produces more valid and sensible results under small sample conditions. Furthermore, to remove the biases, we have conducted Spearman's test as well to look into how sensitively the data is behaving under different tests of the same nature. Figure 1 presents the research methodology for this study.

## RESULTS & DISCUSSION

This section provides the results of both SLRs, both surveys, and both types of tests *i.e.,* parametric, and non-parametric.

### Study selection in GSD context

In the first phase after applying the search strings to different databases, we got 257 results: 53 results from IEEE, 105 from ACM, 40 from Springer, 27 from Science Direct, and 32 from Wiley. In the second step, based on the title and abstract 70 papers were selected. The studies cannot be finalized based on the title and abstract only, so we continued further. In the third step, duplicate papers were removed, and 66 papers were left. Further, based on the full text, we were left with only 28 papers. In the subsequent step, 13 papers were shortlisted primarily based on quality assessment criteria.

In the next step, we carried out the snowballing method (*Wohlin, 2014*) to check the references of the 13 papers. We found 13 papers and applied the same selection process to these papers, and eventually seven papers were selected in this phase. So, after an extensive filtration process, we got 20 final studies. The process is shown in Fig. 2.

### Study selection in in-house context

In the initial part, after applying the search strings to different databases, we got 2,100 results. 415 results from IEEE, 488 from ACM, 569 from Springer, 107 from Science Direct, and 521 from Wiley. In the second step, based on the title and abstract, 111 publications were selected. Within the third step, duplicate papers were removed, and 101 papers were left. Further, based on the full text, we were left with only 38. After this, 17 papers were shortlisted based on quality assessment criteria. In the next step, we applied the snowballing technique (*Wohlin, 2014*) to check the references of the 17 papers. We found 10 papers, applied the same selection process to these papers, and eventually two papers were selected during this process. Therefore, after an intensive filtration process, we obtained 19 final studies. All this process is shown in Fig. 3. The same quality assessment criteria were used for this SLR as well.

### Data extraction and synthesis

Data was carefully extracted from the papers and all authors were involved in this step for the removal of biases. Data synthesis was performed, and we got a list of SPI success factors from 20 studies that were selected for the final stage of SLR performed in the GSD context. Similarly, we got a list of SPI success factors from 19 studies selected during SLR for an in-house context. Initially, we found 35 success factors of SPI related to the GSD environment and 33 SPI success factors related to the in-house development environment.

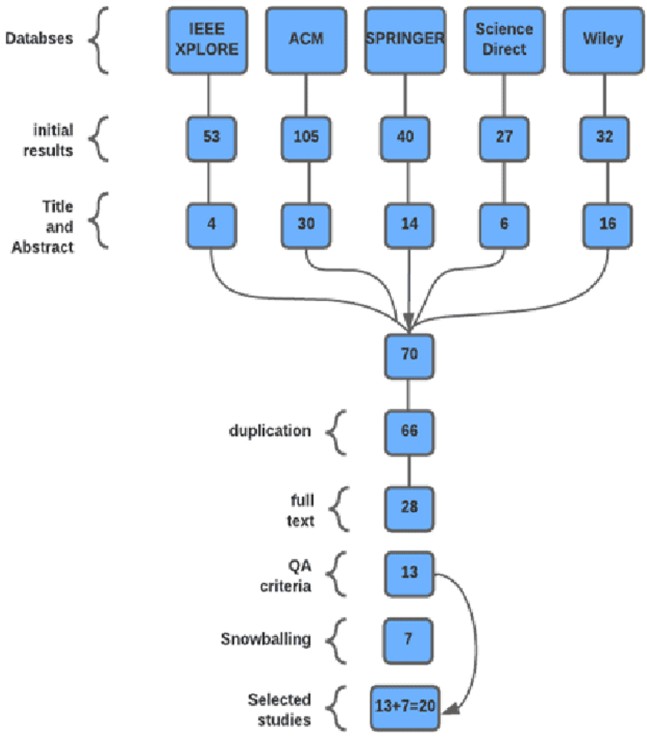

**Figure 2** **SLR for SPI success factors in the GSD environment.**

Table 3 represents the SPI success factors in the GSD environment. The success factors have been represented as $G_1$, $G_2$, $G_3 \ldots G_{35}$.

This completes the answer to RQ1.

Table 4 represents the success factors of SPI in the in-house context. The success factors have been represented as $I_1$, $I_2$, $I_3 \ldots I_{33}$.

This provides the answer to RQ2.

## Results of the first questionnaire survey

This section provides the results of the first survey. From the first questionnaire survey, we got four additional success factors of SPI in the GSD environment, and four success factors of SPI in the in-house development environment.

The additional SPI success factors for GSD context are: On-time delivery, Identification of possible roadblocks, Time management, and Understandable documentation. The additional SPI success factors for in-house context are: understandable documentation, on-time delivery, recreational activities, and self learning. This provides the answer to RQ3.

The additional GSD SPI success factors have been represented as $G_{36}$, $G_{37}$, $G_{38}$, and $G_{39}$ and have been represented in Table 3. The additional in-house SPI success factors have been represented as $I_{34}$, $I_{35}$, $I_{36}$, and $I_{37}$ and have been represented in Table 4.

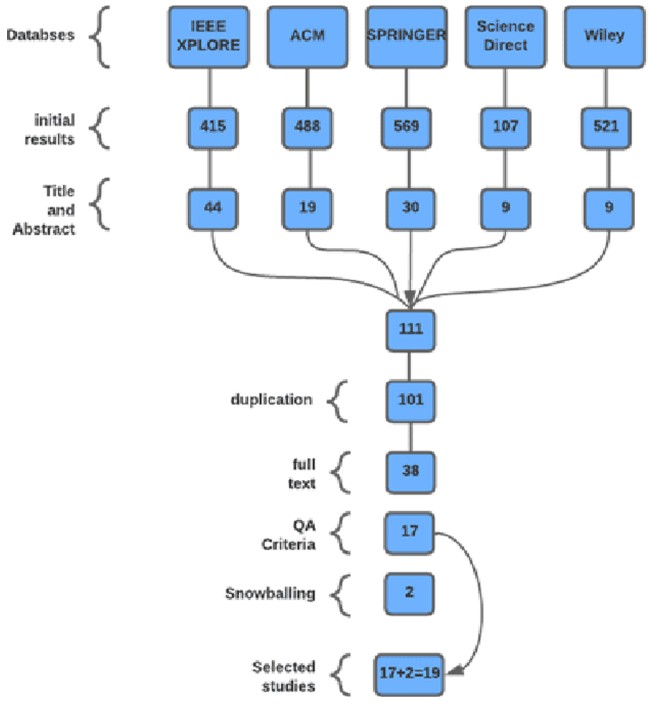

**Figure 3** SLR for SPI success factors in in-house environment.

## Common SPI success factors in GSD and in-house context

By comparing Tables 3 and 4, we have found that there are a total of 17 SPI success factors that are common in both development environments. The common SPI success factors have been represented as $C_1$, $C_2$, $C_3$…$C_{17}$:

$C_1$ -SPI leadership.

$C_2$ -project management.

$C_3$ -Communication.

$C_4$ -Teamwork.

$C_5$ -Setting SPI goals.

$C_6$ -SPI awareness.

$C_7$ -Allocation of resources.

$C_8$ -SPI consultancy.

$C_9$ -Staff involvement.

$C_{10}$ -Senior management commitment and support.

$C_{11}$ -Organizational infrastructure.

$C_{12}$ -Customer involvement/ Client support.

$C_{13}$ -Motivation.

$C_{14}$ -Training.

$C_{15}$ -Reward schemes.

$C_{16}$ -On-time delivery.

$C_{17}$ -Easy to understand documentation.

**Table 3** SPI success factors in the GSD environment.

| IDs | Success factors | IDs | Success factors |
|---|---|---|---|
| $G_1$ | SPI leadership | $G_{21}$ | Joint management infrastructure |
| $G_2$ | Efficient project management | $G_{22}$ | SPI consultancy |
| $G_3$ | Track record of successful projects | $G_{23}$ | Staff involvement |
| $G_4$ | Political stability | $G_{24}$ | Organizational culture |
| $G_5$ | Pilot project performance | $G_{25}$ | Information Sharing |
| $G_6$ | Data protection laws | $G_{26}$ | Senior management commitment and support |
| $G_7$ | SPI certification laws | $G_{27}$ | Organizational infrastructure |
| $G_8$ | Efficient contract management | $G_{28}$ | Overseas site response |
| $G_9$ | Knowledge of clients' language and culture | $G_{29}$ | SPI expertise |
| $G_{10}$ | Continuous organizational support | $G_{30}$ | Effective requirement analysis |
| $G_{11}$ | SPI standards and procedures | $G_{31}$ | Effective customer involvement |
| $G_{12}$ | Mutual understanding among members | $G_{32}$ | Motivation |
| $G_{13}$ | Process improvement evaluation | $G_{33}$ | Training |
| $G_{14}$ | 3Cs (control, communication, coordination) | $G_{34}$ | Risk sharing |
| $G_{15}$ | Skilled human resources | $G_{35}$ | Rich technology infrastructure |
| $G_{16}$ | Setting SPI goals | $G_{36}$ | On time delivery |
| $G_{17}$ | Reward schemes | $G_{37}$ | Identification of possible road blocks |
| $G_{18}$ | Allocation of resources | $G_{38}$ | Time management |
| $G_{19}$ | Trust | $G_{39}$ | Understandable documentation |
| $G_{20}$ | SPI awareness | | |

This provides the answer to RQ4.

The identified common SPI success factors speed up SPI in the case of large-scale companies, where both approaches are followed, and also enable such companies to produce quality software products.

## Results of the second questionnaire survey

Another industrial survey was performed to find out the relative importance of the common SPI success factors. This section provides the results of the second industrial survey and completes the answer to research RQ5.

We have conducted a correlation analysis of the survey results to see if there is any relation between the common SPI success factors of GSD and in-house environments. For this purpose, we have used statistical analysis including the Pearson correlation as a parametric test (*Obilor & Amadi, 2018*) and, Kendall's Tau and the Spearman correlation as non-parametric tests (*Fagerl, 2012*) because the sample is small. Two same types of tests help to cross-examine the result. The sample size is significantly small *i.e.,* 29 total correct responses. However, we were able to evaluate which common SPI success factors are more strongly correlated in GSD or in-house working environments respectively. First, we have applied a parametric test which is the Pearson correlation, the results are shown in Table 5.

We have found that SPI leadership, communication, teamwork, SPI awareness, allocation of resources, SPI consultancy, organizational infrastructure, motivation, on-time delivery,

**Table 4  SPI success factors for in-house development environment.**

| IDs | Success factors | IDs | Success factors |
|-----|-----------------|-----|-----------------|
| $I_1$ | Senior management commitment and support | $I_{20}$ | Customer Involvement |
| $I_2$ | Staff involvement | $I_{21}$ | Skills |
| $I_3$ | Experience of staff | $I_{22}$ | Time allocation |
| $I_4$ | Training | $I_{23}$ | Motivation |
| $I_5$ | Allocation of resources | $I_{24}$ | Project managers personality |
| $I_6$ | Communication | $I_{25}$ | Team leader support |
| $I_7$ | SPI goals | $I_{26}$ | Organizational structure |
| $I_8$ | Tools | $I_{27}$ | SPI consultancy |
| $I_9$ | Reward schemes | $I_{28}$ | Client support |
| $I_{10}$ | Monitoring and feedback | $I_{29}$ | Employee support |
| $I_{11}$ | SPI leadership and procedures | $I_{30}$ | Participation of top leader |
| $I_{12}$ | Teamwork | $I_{31}$ | Automated tools |
| $I_{13}$ | Change management | $I_{32}$ | Tailoring of process |
| $I_{14}$ | Roles and responsibilities | $I_{33}$ | Managing the project |
| $I_{15}$ | SPI personal respect | $I_{34}$ | Understandable documentation |
| $I_{16}$ | Exploitation of existing knowledge | $I_{35}$ | On time delivery |
| $I_{17}$ | SPI awareness | $I_{36}$ | Recreational activities |
| $I_{18}$ | Training and mentoring | $I_{37}$ | Self-learning |
| $I_{19}$ | Exploration of new knowledge | | |

**Table 5  Pearson correlation.**

| Sr. # | Success factors | GSD | In-house |
|-------|-----------------|-----|----------|
| 1 | SPI leadership | 0.265 | 0.045 |
| 2 | Project management | 0.178 | 0.219 |
| 3 | Communication | 0.306 | 0.088 |
| 4 | Teamwork | 0.355 | 0.229 |
| 5 | Setting SPI goals | 0.097 | 0.130 |
| 6 | SPI awareness | 0.105 | 0.076 |
| 7 | Allocation of resources | 0.457 | 0.111 |
| 8 | SPI consultancy | 0.240 | 0.132 |
| 9 | Staff involvement | 0.059 | 0.170 |
| 10 | Senior management commitment and support | 0.219 | 0.357 |
| 11 | Organization infrastructure | 0.129 | 0.018 |
| 12 | Customer involvement/ Client support | 0.053 | 0.097 |
| 13 | Motivation | 0.212 | 0.043 |
| 14 | Training | 0.193 | 0.327 |
| 15 | Reward scheme | 0.264 | 0.292 |
| 16 | On time delivery | 0.267 | 0.237 |
| 17 | Easy to understand documentation | 0.182 | 0.005 |

| Table 6 | Kendall's Tau correlation. | | |
|---|---|---|---|
| Sr. # | Success factors | GSD | In-house |
| 1 | SPI leadership | 0.180 | 0.015 |
| 2 | Project management | 0.169 | 0.170 |
| 3 | Communication | 0.246 | 0.019 |
| 4 | Teamwork | 0.309 | 0.210 |
| 5 | Setting SPI goals | 0.012 | 0.098 |
| 6 | SPI awareness | 0.097 | 0.015 |
| 7 | Allocation of resources | 0.325 | 0.076 |
| 8 | SPI consultancy | 0.200 | 0.023 |
| 9 | Staff involvement | 0.035 | 0.086 |
| 10 | Senior management commitment and support | 0.125 | 0.235 |
| 11 | Organization infrastructure | 0.091 | 0.014 |
| 12 | Customer involvement/ Client support | 0.027 | 0.072 |
| 13 | Motivation | 0.109 | 0.061 |
| 14 | Training | 0.112 | 0.282 |
| 15 | Reward scheme | 0.143 | 0.208 |
| 16 | On time delivery | 0.175 | 0.165 |
| 17 | Easy to understand documentation | 0.105 | 0.035 |

and easy to understand documentation are more important in GSD as compared to in-house, whereas, project management, setting SPI goals, staff involvement, senior management commitment and support, customer involvement/client support, training, and reward schemes are more important in the in-house environment as compared to the GSD. To validate the results, we also performed non-parametric tests (Kendall's Tau and the Spearman correlation), and the results were the same as shown in Tables 6 and 7. Table 6 presents the results of Kendall's Tau correlation. Table 7 presents the results of the Spearman correlation.

Using Table 5, the SPI success factors that are more important in the GSD environment as compared to the in-house environment can be arranged in descending order, based on respective scores, as shown in Table 8.

So according to the Pearson correlation (Table 5) the top three SPI success factors, for the GSD environment in the case of the large-scale development organization, with respective scores are: allocation of resources (0.457), teamwork (0.355), and communication (0.306). The results can be validated from Tables 6 and 7 as the results are the same in the case of Kendall's Tau correlation (Table 6) and the Spearman correlation (Table 7).

According to Kendall's Tau correlation (Table 6), the top three SPI success factors, for the GSD environment, with respective scores are: allocation of resources (0.325), teamwork (0.309), and communication (0.246). According to the Spearman correlation (Table 7), the top three SPI success factors, for the GSD environment, with respective scores are: allocation of resources (0.401), teamwork (0.381), and communication (0.294).

**Table 7  Spearman correlation.**

| Sr. # | Success factors | GSD | in-house |
|---|---|---|---|
| 1 | SPI leadership | 0.223 | 0.038 |
| 2 | Project management | 0.213 | 0.219 |
| 3 | Communication | 0.295 | 0.019 |
| 4 | Teamwork | 0.381 | 0.259 |
| 5 | Setting SPI goals | 0.031 | 0.132 |
| 6 | SPI awareness | 0.104 | 0.030 |
| 7 | Allocation of resources | 0.401 | 0.091 |
| 8 | SPI consultancy | 0.233 | 0.056 |
| 9 | Staff involvement | 0.039 | 0.126 |
| 10 | Senior management commitment and support | 0.162 | 0.317 |
| 11 | Organization infrastructure | 0.115 | 0.013 |
| 12 | Customer involvement/ Client support | 0.028 | 0.089 |
| 13 | Motivation | 0.134 | 0.070 |
| 14 | Training | 0.144 | 0.401 |
| 15 | Reward scheme | 0.206 | 0.274 |
| 16 | On time delivery | 0.232 | 0.219 |
| 17 | Easy to understand documentation | 0.145 | 0.045 |

**Table 8  Important SPI success factors in GSD.**

| Sr. # | Success factors | GSD score |
|---|---|---|
| 1 | Allocation of resources | 0.457 |
| 2 | Teamwork | 0.355 |
| 3 | Communication | 0.306 |
| 4 | On time delivery | 0.267 |
| 5 | SPI leadership | 0.265 |
| 6 | SPI consultancy | 0.240 |
| 7 | Motivation | 0.212 |
| 8 | Easy to understand documentation | 0.182 |
| 9 | Organization infrastructure | 0.129 |
| 10 | SPI awareness | 0.105 |

Again, using Table 5, the SPI success factors that are more important in the in-house environment as compared to the GSD environment can be arranged in descending order, based on respective scores, as shown in Table 9.

According to the Pearson correlation (Table 5), the top three SPI success factors, for the in-house environment in the case of large-scale software development organizations, with respective scores are: senior management commitment and support (0.357), training (0.327) and reward scheme (0.292). The results can be validated from Tables 6 and 7 as the results are the same in the case of Kendall's Tau correlation (Table 6) and the Spearman correlation (Table 7).

**Table 9  Important SPI success factors for in-house environment.**

| Sr. # | Success factors | in-house score |
| --- | --- | --- |
| 1 | Senior management commitment and support | 0.357 |
| 2 | Training | 0.327 |
| 3 | Reward scheme | 0.292 |
| 4 | Project management | 0.219 |
| 5 | Staff involvement | 0.170 |
| 6 | Setting SPI goals | 0.130 |
| 7 | Customer involvement/client support | 0.097 |

According to the Kendall's Tau correlation (Table 6), the top three SPI success factors, for the in-house environment, with respective scores are: training (0.282), senior management commitment and support (0.235), and reward scheme (0.208). According to the Spearman correlation (Table 7), the top three SPI success factors, for the in-house environment, with respective scores are: training (0.401), senior management commitment and support (0.317), and reward scheme (0.274).

# THREAT TO VALIDITY

The possible threats to the validity of this study are presented as:

## Construct validity

The study is based on 35 SPI success factors in the GSD context and 33 SPI success factors in the in-house environment for large-scale software development companies. In both cases, SPI success factors have been identified through SLR by following the recommended steps. The lists of the SPI success factors have been provided to respondents in both cases. Furthermore, to improve readability and understanding, both questionnaires were tested through pilot studies. Based on the results of the pilot study, the final versions of the questionnaire surveys were developed. Therefore, the participants were familiar with the SPI success factors.

## Internal validity

The participants have working experience of 1 to 5 years and they belong to such large software development companies where GSD and in-house approaches are followed. Therefore, the SPI success factors were associated with the working place of the participants. To examine the relative importance of the common SPI success factors in both environments, a 5-point Likert scale has been employed that has already been used in many studies.

## External validity

To conduct the study, a sample of 50 professionals has been selected from 50 large-scale software development companies where GSD and in-house approaches are practiced. This is quite possible that all the corresponding practitioners do not entirely agree with the results of this study, but the sample is truly illustrative of the population. As out of 110

companies, 50 were chosen for the study based on size, development approaches being followed, and relevant available respondents. Moreover, to reveal which common SPI success factor carries more value in which development environment and to validate the results, the Pearson correlation (parametric), Kendall's Tau, and the Spearman correlation (non-parametric) have been calculated.

## CONCLUSIONS

This study focuses on the identification of SPI success factors for in-house and GSD in the case of large-scale software development organizations. We have identified 35 SPI success factors for GSD through the first systematic literature review (SLR). This answers RQ1. Similarly, 33 SPI success factors have been identified in the in-house environment through a second SLR. This answers RQ2. A survey has been performed to explore additional SPI success factors from the industry for both environments. We have found four additional SPI success factors in the in-house environment and also four additional SPI success factors in the GSD environment. This answers RQ3. Thus, a comprehensive list of SPI success factors, for both environments, in the case of large-scale software development organizations, contains 39 SPI success factors for the GSD and 37 SPI success factors for the in-house environment. By comparing SPI success factors for both environments, we have found 17 common SPI success factors. This answers RQ4. The common SPI success factors accelerate and promote the SPI in the case of such large-scale software development organizations that operate in both environments.

To explore which common SPI success factor is more important out of both environments, a second industry survey has been conducted. To analyze and validate the results, the Pearson correlation, Kendall's Tau and the Spearman correlation have been performed. This answers RQ5. This has been revealed that SPI leadership, communication, teamwork, SPI awareness, allocation of resources, SPI consultancy, organizational infrastructure, motivation, on-time delivery, and easy to understand documentation are more valuable SPI success factors in the case of the GSD as compared to the in-house environment. Out of these 10 SPI success factors, the top three SPI success factors for GSD are allocation of resources, teamwork, and communication.

Furthermore, project management, setting SPI goals, staff involvement, senior management commitment and support, customer involvement/client support, training, and reward schemes are more valuable SPI success factors for the in-house environment as compared to GSD. Out of these seven SPI success factors, the top three SPI success factors for the in-house environment are senior management commitment and support, training, and reward scheme.

### Funding

This research work has been supported by Umm AL-Qura University, Al-lith, Saudi Arabia. The funders had a role in study design, data collection and analysis, and preparation of mansucript. The funders had no role in the decision to publish.

## Grant Disclosures

The following grant information was disclosed by the authors:
Umm AL-Qura University, Al-lith, Saudi Arabia.

## Competing Interests

The authors declare there are no competing interests.

## Author Contributions

- Javed Iqbal conceived and designed the experiments, analyzed the data, performed the computation work, prepared figures and/or tables, and approved the final draft.
- Haris Jibran conceived and designed the experiments, performed the computation work, prepared figures and/or tables, and approved the final draft.
- Ahmad Sami Al-Shamayleh conceived and designed the experiments, performed the experiments, prepared figures and/or tables, and approved the final draft.
- Fakhar Abbas performed the experiments, prepared figures and/or tables, and approved the final draft.
- Adnan Akhunzada performed the experiments, performed the computation work, authored or reviewed drafts of the article, and approved the final draft.
- Salman Z. Alharthi conceived and designed the experiments, analyzed the data, performed the computation work, prepared figures and/or tables, authored or reviewed drafts of the article, and approved the final draft.
- Abdullah Gani analyzed the data, authored or reviewed drafts of the article, and approved the final draft.

## Data Availability

The raw data is available in the Supplemental Files.

## Supplemental Information

Supplemental information for this article can be found online at http://dx.doi.org/10.7717/peerj-cs.1656#supplemental-information.

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
