# Peer review of "Comparative study of SPI success factors in global and in-house environment for large-scale software companies"

_PeerJ Computer Science, doi:10.7717/peerj-cs.1656_

## Round 0.1 · original submission · Major Revisions

The reviewers suggested some substantial revisions, which I hope you will consider and address. Once that is done, I would like to invite you to resubmit your manuscript for further consideration by the reviewers.
Remember to highlight the revision parts with colors in the revised manuscript, together with a point-by-point response letter.

The comments in detail provided by reviewers are given below.

Reviewer 1 ·

Basic reporting

The authors have identified the success factors of SPI for both development approaches (GSD and In-house) in the case of large-scale software companies. Methods. Two systematic literature reviews
have been performed. An industrial survey has been conducted to detect additional SPI
success factors for both development environments.
the systematic literature reviews were not documented properly. The results and discussion parts were missing. the RQs of SLRs would be answered first, then the survey results should be presented, and a combined discussion should be made at the end. The paper looks incomplete in its present form and I believe the authors need to add more details of all the hard work they have done in conducting two SLRs and an industrial survey.
Questionnaire validation needs to be explained in detail.

Experimental design

The SLRs should be properly explained, with all of the RQs answered and discussed.
the industrial survey needs to be justified. The questionnaire used for the survey needs to be validated, and details should be added to the paper. In addition, the authors should justify the need to conducting two SLRs rather than one with multiple RQs.

Validity of the findings

the findings should be supported throughout the process of conducting this research which is missing in the paper. the SLRs need to be explained in detail and so is the questionnaire survey.

Additional comments

I suggest the authors to make necessary changes to the paper to make it a beneficial reading for the audiances.

Reviewer 2 ·

Basic reporting

Satisfactory

Experimental design

Satisfactory

Validity of the findings

Satisfactory

Additional comments

The authors have made reasonable effort, but the paper requires certain additions for improvement. Some recommendations for enhancement are:
1): Please revise the paper thoroughly to remove typographical errors. For example:
a): Most of the times authors have used the word In-house but in the Introduction section line number 99 contains the word in-house. Change it too to In-house.
b): In the Methodology section, Part B, line number 252, replace section one with Section-I as the other two sections have been mentioned, section-II and section-III.
Check the whole manuscript for such inconsistencies.
2): Please remove the abbreviations used in research questions like GSD, SPI.
3): For the first questionnaire survey, authors have not mentioned that it is a Cross sectional study or Longitudinal study. Please mention it.
4): Similarly, For the second questionnaire survey, authors have not mentioned that it is a Cross sectional study or Longitudinal study. Please mention it.
5): In the Methodology Section, Part B that is Data Collection Via Questionnaire Surveys, the details about the first questionnaire survey does not mention that there are how many factors in the list of SPI success factors in the case of GSD. Same is the case for In-house development. Please mention the relevant number in both cases.
6): In the Methodology Section, Part B, the details about the questionnaire survey employed for second survey does not provide information about the Likert Scale. Please provide details about the Likert Scale like how many points etc.

Reviewer 3 ·

Basic reporting

The study aims to identify the common SPI success factors for GSD and In-house software development in the case of large-scale software development organizations. The topic is important as there is a need to investigate the process-related success factors for a hybrid environment (i.e., both for in-house development and GSD). However, the manuscript lacks motivation for why the investigations presented are essential and what value they will add to research and practice knowledge. Further, the article has some structural issues as well. The detailed review comments are given below in the additional comments.

Experimental design

The methodology section needs some revisions. Please see the details in the additional comments.

Validity of the findings

Threats to the validity of the paper need to be included in the paper.

Additional comments

1. Please motivate the need to investigate the SPI success factors for both environments (i.e., in-house development and GSD). It is essential to highlight the contributions of this study from the perspectives of research and practice. Please use valid references while presenting the need for your work.
2. Use related work to support your investigation. The argument presented in the paper of not having any study on the topic is not enough reason to conduct such an extensive investigation comprising two SLRs and two surveys.
3. Threats to the validity of this study need to be included in the paper.
4. Some methodology steps are merged in the results & discussion section. I suggest moving parts of subsections (A, B, C, D, E, F & H) presenting study execution procedures should be moved to the methodology section. The results & discussion section should only contain the study results and the researchers’ reflections as a discussion.
5. Further, providing details about the systematic literature review (subsection A) in the Methodology section, it has been claimed that the protocol step contains five elements. The details of element one have already been provided, but details of the four elements have been provided “in this section.” Please explain or mention what you mean by “this section.”
6. In subsections F and H (Results and Discussion), it has been stated that the Stratified sampling method has been employed. But details of the sampling method are missing that must be provided.
7. In subsections F, SAMPLING AND POPULATION (Results and Discussion), provide details about the respondents of the first questionnaire survey, like experience, job nature, etc.
8. Also, subsections H, SAMPLING AND POPULATION (Results and Discussion), provide details about the respondents of the second questionnaire survey.
9. In the Conclusions, associate the results with the respective research questions, i.e., RQ1, RQ2, RQ3, RQ4, and RQ5.

---

## Round 0.2 · accepted · Accept

The authors have addressed all the comments from the reviewers.

Reviewer 1 ·

Basic reporting

The authors have successfully demonstrated all the suggested changes.

Experimental design

The authors have successfully demonstrated all the suggested changes.

Validity of the findings

The authors have successfully demonstrated all the suggested changes.

Additional comments

The authors have successfully demonstrated all the suggested changes.

Reviewer 2 ·

Basic reporting

The content and presentation of the paper has been improved.

Experimental design

ok

Validity of the findings

The contributions are clear and easy to follow.

Additional comments

All my concerns have been addressed.

Reviewer 3 ·

Basic reporting

The authors have addressed most of my comments. Therefore, I have no more suggestions. However, I strongly recommend that the authors carefully review the paper for grammar and typos and make necessary adjustments.

Experimental design

I do not have any further suggestions.

Validity of the findings

I do not have any further suggestions.

Additional comments

No comments.